# Role of Information in Farmers' Response to Weather and Water Related Stresses in the Lower Bengal Delta, Bangladesh

**Uthpal Kumar** [1,*] , **Saskia Werners** [1], **Sharmishtha Roy** [2], **Sadia Ashraf** [2], **Long Phi Hoang** [1], **Dilip Kumar Datta** [2] **and Fulco Ludwig** [1]

[1]  Water Systems and Global Change, Wageningen University and Research, P.O. Box 47, 6700 AA Wageningen, The Netherlands; saskia.werners@wur.nl (S.W.); long.hp2002@gmail.com (L.P.H.); fulco.ludwig@wur.nl (F.L.)

[2]  Environmental Science Discipline, Life Science School, Khulna University, 9208 Khulna, Bangladesh; sharmi.royes@gmail.com (S.R.); sadiaes11@gmail.com (S.A.); dkd_195709@yahoo.com (D.K.D.)

\*  Correspondence: uthpal.kumar@wur.nl

**Abstract:** Farmers in the lower Bengal Delta around the city of Khulna, Bangladesh, are particularly vulnerable to hydro-climatic variability. Phenomena such as heavy rain, drought and salt intrusion increasingly affect their crop production, with far-reaching socio-economic and environmental impacts. Reliable hydro-climatic information service received in a timely manner could help farmers improve their responses to hydro-climatic variability, thus improving their agricultural decision-making. However, significant challenges persist regarding information uptake and the role of information from the available sources. We designed an explorative research framework combining different participatory methods and analysis of climate data. Our aim was to examine three key research questions: (i) what information is currently available to farmers for agricultural practices and decision-making? (ii) what is the perceived quality of the available hydro-climatic information in response to water and weather related stresses? (iii) how does the available information influence farmers' decision-making? We found that farmers had access to information from five main sources: informal contacts, formal contacts, education and training programs, traditional media (like television) and modern ICT tools/social media. However, informal contacts, particularly with peer farmers and private input suppliers, were the farmers' main source, in addition to their own previous experiences. Farmers perceived hydro-climatic variability as high and the quality of available hydro-climatic information as poor. They indicated a need for more accurate, time-specific, trusted and actionable information for improving agricultural decision-making. We conclude that there is high potential and need for hydro-climatic information services tailored for farmers in the study area.

**Keywords:** hydro-climatic information; agricultural decisions; Bengal Delta; Bangladesh

## 1. Introduction

Agriculture is the backbone of the rural economy and livelihoods in Bangladesh. The sector contributes over 15% of the national GDP and generates income and employment for some 43% of the population [1,2]. The Lower Bengal Delta, in south-west Bangladesh, is an ecologically rich and highly productive agricultural zone. Farming in the delta, however, is vulnerable to hydro-climatic variability [3–6]. Here we understand vulnerability as the state of susceptibility to harm from water and weather-related hazards such as drought, cyclone, heavy or reduced rainfall, flood, etc., and the hydro-climatic variability is the short-term changes or long-term shifts of the water (availability, quality, quantity and timing, etc.) and weather (temperature, rainfall, wind, etc.) phenomena that have an impact on the overall agricultural production

system in a particular geographical area [7–9]. Agriculture in the delta has become increasingly difficult and risky due to the greater unpredictability of rainfall [10,11]. As a result, farmers increasingly confront problems such as severe waterlogging, salinity intrusion and drought [9,12,13]. Moreover, these problems disproportionately affect poor farmers and smallholders [3,8,11].

The literature suggests that reliable hydro-climatic information received in a timely manner would help farmers improve their responses to hydro-climatic variability, leading to improved agricultural decision-making [14–16]. However, significant challenges persist regarding information uptake and the role of information from the available sources [17–19]. We approached smallholder farmers and initiated discussions on how they accessed weather and water-related information, such as seasonal weather forecasts, rainfall trends, temperature stresses (hot and cold spells), water and soil salinity and cyclonic (storm-related) weather emergencies; and what was the perceived quality and role of information found from the different sources in managing hydro-climatic variability.

Indeed, information is acknowledged as a key agricultural input [20,21]. Farmers draw on many sources of information when considering ways to reduce risks and production uncertainties [22–24]. Information on hydro-climatic events is vital for both strategic and tactical decision-making [25]. Strategic decisions are those that concern the long term, while tactical decisions concern steps that can be taken in the short to medium term. In both types of decisions, farmers draw on information regarding hydro-climatic conditions [25]. For example, seasonal forecasts can improve strategic decisions on, for example, crop types and variety selection, and they can help farmers prepare for weather and water hazards [26]. Mid-term forecasts (2–4 weeks in advance) can inform farmers' tactical decisions, such as optimization of planting and harvesting dates. Short-term forecasts can help in day-to-day decisions, such as regarding livestock evacuation, crop protection and storage, and management of household and farm assets [27].

In the Lower Bengal Delta, farming communities base agricultural decision-making mainly on traditional practices [27]. These draw on farmers' own experiences, as well as practical knowledge passed between them and from generation to generation. Bangladesh's Department of Agricultural Extension (DAE) also uses a traditional approach to provide extension information to farmers [28]. The DAE's main communication platforms are personal and group contacts and traditional media, such as radio, television and printed manuals. Research shows that the majority of the field extension workers have limited access, usage, knowledge and capacity for ICT-led extension services [29]. Besides, farmers also have several limitations to accessing ICT facilities, inadequate information services, and information quality for agricultural decision-making [2,30]. However, particularly in today's context of accelerating climate change and hydro-climatic variability, these traditional approaches may be insufficient to inform farmers adequately and on time in managing hydro-climatic risks [2,17]. Currently, there is a bridge-gap between the hydro-climatic information producer such as the Bangladesh Meteorological Department (BMD), the DAE as a service intermediary, and end-user farmers. The hydro-climatic information services thus required urgent inclusion in the existing agricultural knowledge and information systems for linking farmers and the information producer for shared learning, dissemination, and better-informed agricultural decision-making [17]. The hydro-climatic information services are still on an emergency basis and for the regional-scale information dissemination that has less usability for local communities [17]. However, to manage new climate and weather-related risks, farmers need location- and time-specific information and frequent access to local extension officers [21,31]. Finally, the country's main sources of hydro-climatic information belong to the Bangladesh Meteorological Department (BMD) and the Bangladesh Water Development Board (BWDB), and their reports are hardly disseminated to farmers for agricultural decision-making purposes [17,32].

A major challenge, therefore, is to find ways to get hydro-climatic information to local farmers in formats that are useful to support their decision-making. New kinds of information support, technology, organization and expertise are needed to help the vulnerable communities [33]. Another key challenge is to tailor hydro-climatic information for local farmers and get it to these farmers enough in advance that it can inform strategic and tactical decisions on agricultural practices. In this regard, co-production

can be an ideal approach [34,35]. Indeed, co-production studies are currently narrowly framed [36]; and several challenges exist in the socio-economic, socio-political and cultural contexts [37,38]. Thus, to effectively tailor and communicate information to farmers in a local context, a detailed understanding of farming practices is needed [39]. Vaughan and Dessai [40] found a mismatch between information needs and information being provided to the sectoral users. Knowledge is required, for instance, on the time horizon in which key decisions are made [41]. In addition, it is important to know what information sources are now available to farmers, and how farmers perceive the quality of information from these sources, as well as their perception of the value of existing platforms in providing useful information.

With this in mind, we posed three research questions centered on hydro-climatic information services for farmers in the delta: (i) What information is currently available to farmers to inform their agricultural practices and decision-making? (ii) To what extent do farmers perceive the available hydro-climatic information as helpful in responding to water- and weather-related stresses? (iii) How has the available information influenced farmers' decision-making? To answer these questions, we designed an exploratory research framework combining field visits, farmer interviews, focus group discussions and expert interviews with analysis of climate data. Our aim was to map and understand the information sources available in the study area, to identify the limitations of the existing sources, and to suggest ways to better design information for increased uptake at the local level.

## 2. Materials and Methods

### 2.1. Description of the Study Sites

The Bengal Delta, also known as the Ganges-Brahmaputra-Meghna (GBM) Delta, located in the northern shores of the Bay of Bengal, is one of the populous deltas of the world [42,43]. This delta has a unique ecosystem characteristic comprising the three mighty river systems (Ganges-Brahmaputra-Meghna) enclosed by the terrestrial, aquatic and marine ecosystems [44]. The city Khulna is located in the lower part of the Bengal Delta frequently stressed by tidal surge related inundation, salinity intrusion, tropical cyclone and hydro-climatic variabilities [44–48]. The city is highly dependent on peri-urban agriculture for its food supply. Khulna is also a regional food supply hub, though this region of Bangladesh is particularly vulnerable to the effects of climate change. The city's importance in regional food supply and the vulnerability of its farming communities to climate change impacts were key reasons for its selection for this study. Indeed, Khulna is a zone of multiple vulnerabilities as well as opportunities [3,49]. It is the third largest (64.78 km$^2$) metropolitan area in Bangladesh. Khulna district has 9 upazilas (sub-districts) and about 2.3 million inhabitants [50].

The climate in Khulna is subtropical warm and humid, with four distinct seasons: (i) dry winter (December to February), (ii) hot pre-monsoon summer (March to May), (iii) rainy monsoon (June to September) and (iv) post-monsoon autumn (October to November) [51].

We obtained climatological data from the Bangladesh Meteorological Department (BMD). According to this data, over the 1948–2018 period the average annual rainfall was 1752.3 mm and the mean annual air temperature was 26.7 °C (see Supplementary Material A). The average monthly minimum and maximum temperatures were 21.9 and 31.3 °C, respectively. January was the coldest month, with a mean minimum temperature of 12.9 °C. April was the warmest month, with a mean maximum temperature of 34.9 °C, with an average annual rainfall anomaly of ±48.6% to 35.5% from 1981 to 2014. According to the literature, some 80% of precipitation occurs in the monsoon season from May to September [51–53]. However, our data for 1948–2018 indicate that the monsoon season extended from May to October, with some 90% of total annual rainfall occurring during these months (see Supplementary Material A). The highest and lowest rainfall quantities were found in July and December, with monthly averages of 327.6 mm and 4.5 mm, respectively. Due to the abundance of rainfall in the region, the area offers excellent opportunities for rainfed agriculture [54].

For our study, we selected two sub-districts: Batiaghata (~248 km$^2$) and Rupsa (~120 km$^2$) (Figure 1). Major crops grown here were paddy, jute, sesame and vegetables, with small-scale aquaculture-agriculture

also observed year-round. Farmers grew various short-term crops and vegetables as well, such as beans, gourds, eggplants and tomatoes, in integrated aquaculture-agriculture farming systems. Integrated farming systems have been found to provide greater economic returns than paddy or vegetable monocrops. Additionally, [55] found that an integrated farming system consisting of paddy, vegetables and aquaculture was more resilient to recurrent hydro-climatic variability.

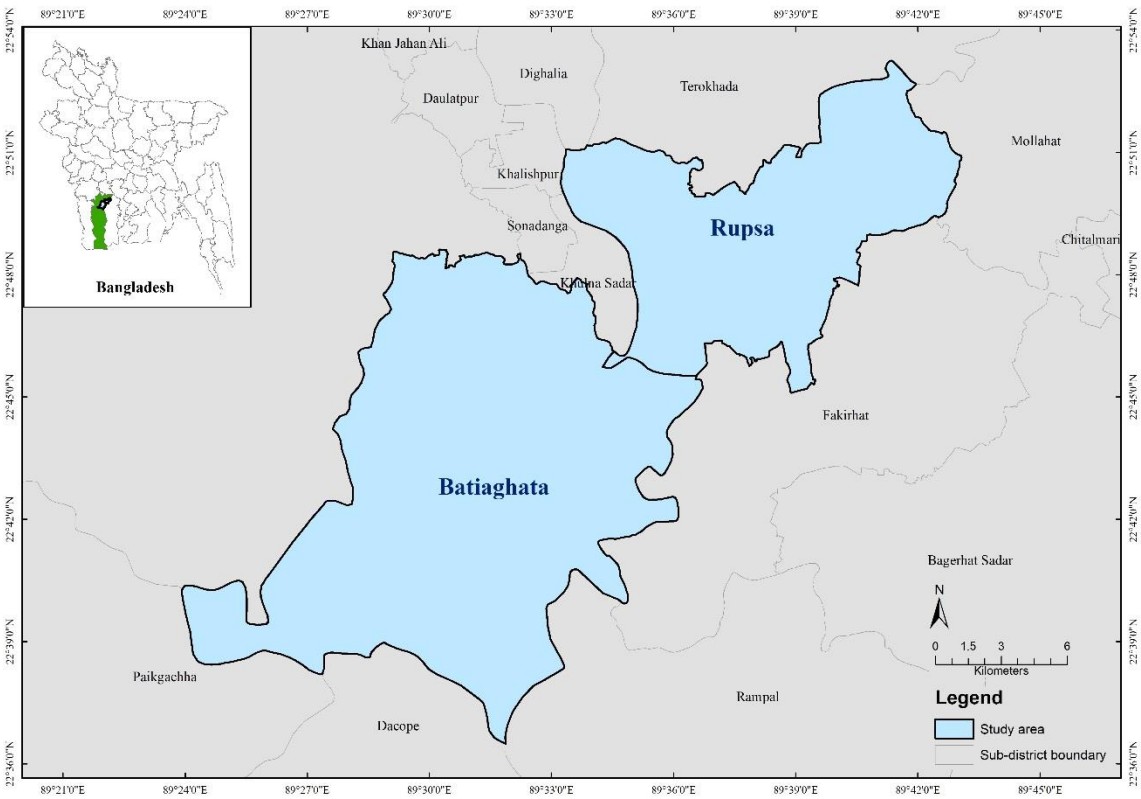

**Figure 1.** Map showing location of the study area in Khulna, Bangladesh. Khulna is indicated on the inset map in green. The two study sites, Batiaghata and Rupsa, are in light blue. The city of Khulna is located between Batiaghata and Rupsa.

## 2.2. Site Selection, Data Collection and Analysis

This study used an explorative research framework combining desk research, secondary data collection, field visits, focus groups and expert interviews (Figure 2). A mixed-method approach for this study allows us to collect information and triangulation through different participatory tools and approaches.

We focused on six peri-urban villages within Rupsa and Batiaghata sub-districts, where we carried out 200 farmer interviews, 4 focus group discussions and 20 expert interviews from August 2017 to May 2018. Questionnaires (Supplementary Material B) and checklists (Supplementary Materials C and D) guided primary data collection through the interviews and focus groups. Table 1 describes the stakeholders engaged and methods of data collection applied for each. Secondary data were collected from documentary sources: journals, reports from the Bangladesh Bureau of Statistics (BBS) and unpublished documents from DAE district and sub-district offices regarding crops, population and livelihood characteristics of farm households in the study area.

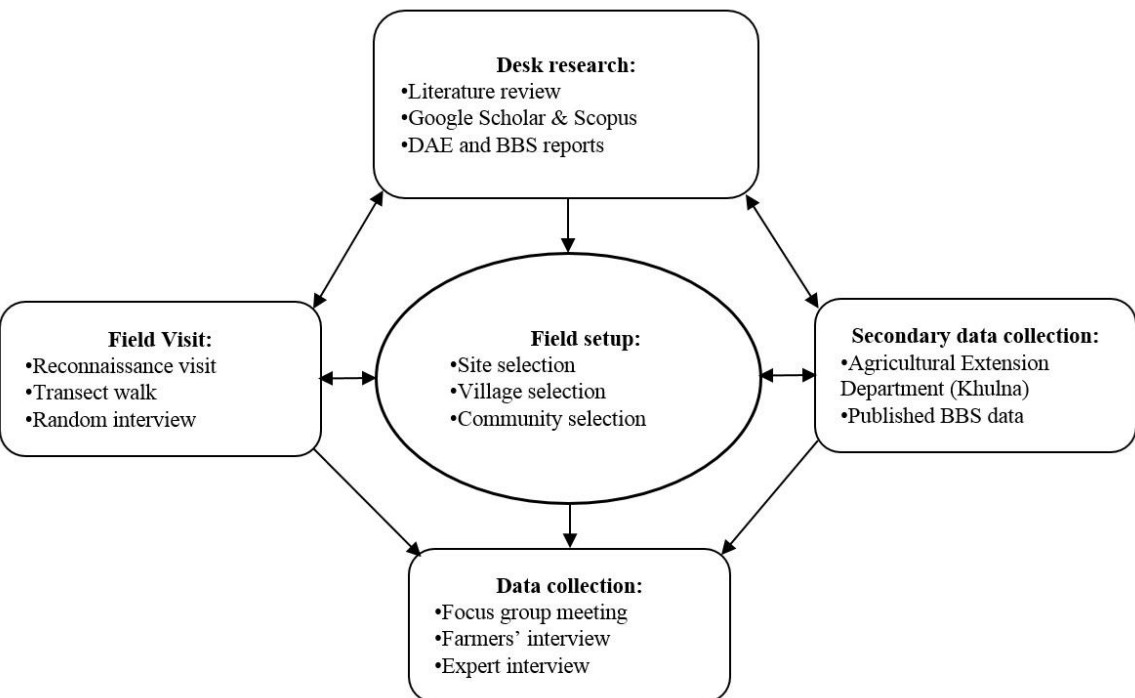

**Figure 2.** Study and data collection framework.

**Table 1.** Stakeholders and methods of primary data collection.

| Stakeholders | Participants | Methods/Tools | Reason for Engagement |
|---|---|---|---|
| Local farmers | Men (138)<br>Women (62) | - Field visits<br>- Transect walk<br>- Farmer interviews<br>- Focus group discussions | Source of primary data on demography, agricultural practices, agricultural information needs and future interest in hydro-climatic information services. |
| Extension officer (district level) | - Deputy Director (1)<br>- District Training Officer (1) | - Consultation meeting<br>- Expert interviews | In charge of extension services at the district level. The district training officer is in charge of agricultural training at the district level. |
| Extension officer (sub-district level) | - Upazila Agriculture Officer (2)<br>- Upazila Agriculture Extension Officer (3) | - Expert interviews<br>- Focus group discussion | Responsible for providing extension services at the sub-district level. |
| Extension officer (field level) | - Sub-Assistant Agriculture Officer (4) | - Expert interviews<br>- Focus group discussion | Provides field-level extension services to farmers, usually grouped as an agricultural block. |
| AIS officer (regional level) | - Regional Farm Broadcasting Officer (1) | - Expert interview | Provides agricultural information at the regional level using traditional and ICT platforms. |
| Met officer (district level) | - Assistant meteorologist (1)<br>- Met assistant (1) | - Expert interviews | Collects daily and hourly data on local weather parameters. |
| Input dealers (local level) | - Local input dealers and retailor (4) | - Expert interviews | Sells inputs to local farmers and provides crop advisory services. |
| Researchers | - Agrotechnology faculty, Khulna University (2) | - Expert interviews | Involved in agricultural extension research for more than 10 years. |
| Total | - Farmers (200)<br>- Experts (20) | | |

Three of the surveyed villages—Sreefaltala, Domra and Peyara—were located in Rupsa, and the remaining three villages—Jharbhanga, Raingamari and Sanchibunia—were in Batiaghata. In Bangladesh, the lowest administrative jurisdiction of government is called a "union". Unions are divided into agricultural blocks, each of which has its own Sub-Assistant Agriculture Officer (SAAO). The SAAO, working under the DAE, provides extension services to some 2000 farm households. All of the selected villages exhibited peri-urban characteristics; that is, they were located in close proximity to the Khulna metropolitan area, and were highly interdependent and interconnected with the city.

After site selection, data on information use in agricultural decision-making were gathered through focus group discussions and farmer and expert interviews, guided by checklists and questionnaires.

*Focus group discussions.* Four focus group discussions were held in Rupsa and Batiaghata. Two meetings were arranged in each sub-district. All the meetings were guided by the same checklist (Supplementary Material B), covering four key topics: (i) current cropping practices, (ii) access to weather and water related information for agricultural decision-making, (iii) traditional knowledge and farming practices and (iv) measures to address weather and water challenges. The focus groups also touched upon key agricultural decisions and time horizons for taking specific decisions related to crops and livelihoods. The focus group meetings were conducted in the farmers' villages in October and November 2017 and involved 10 and 16 participants, respectively, in Rupsa and Batiaghata. Two research assistants took notes during these meetings. Qualitative data from the meetings were summarized and entered into Excel for further analysis and interpretation.

*Farmer interviews.* Two-hundred farmers were interviewed. This sample size was determined following Berenson and Levine (1992) to obtain a 95% confidence level based on the total population of households (858) [13]. In total, 62 households were selected from Batiaghata and 138 from Rupsa. Table 2 presents key data on the interviewed farmers. With the interviews we sought to obtain a quantitative overview of the farming communities and farming practices, including farmers' access to and the quality of information sources and their interest in and need for expanded information services for agricultural decision-making. We designed a semi-structured questionnaire to guide the farmer interviews (Supplementary Material B). These were informed by consultations with experts, field visits and random farmer interviews carried out ahead of time. Furthermore, the questionnaire was pre-tested with 10 randomly selected farmers at both sites. Based on the pre-test, we made minor changes to the questionnaire. Farmer interviews were conducted using the open access online interview tool KoBoToolbox (www.kobotoolbox.org). Two female master's degree students conducted the interviews. In Bangladeshi society, women have easier access to unknown households than men. It took two months to complete the interviews. Simple random sampling was used to select interviewees. Finally, SPSS Statistics 20 software was used to analyze and interpret the quantitative data obtained.

*Expert interviews.* We interviewed 20 experts using open-ended questionnaires (Supplementary Material C, see list of experts in Table 1). Interviews focused on the extension services currently provided, limitations of existing information and extension services, need for hydro-climatic information for agricultural decision-making and current agricultural decision-making practices of farmers in Khulna. Several more specific topics were also discussed with the experts, such as how farmers dealt with hydro-climatic information and what role available information sources and quality played in agricultural decision-making. Finally, the interview results were summarized and entered into Excel for analysis and interpretation.

## 3. Results

### 3.1. Characteristics of Farmers

In this study, 69% of the 200 respondent farmers were from Rupsa and 31% were from Batiaghata. The majority of the farmers were sharecroppers (74%), and about a third (31%) were functionally landless (~0.02 ha) (Table 2). Some 26% of the farmers were illiterate. Only 10% were in the 18–25

age range, indicating that few young men and women were involved in agriculture in the study area. However, a significant number of retired government officials were involved in agriculture at the study sites. All of the interviewed farmers expressed an interest in hydro-climatic information to inform agricultural practices, and 91% expressed interest in a mobile phone application for receiving hydro-climatic information.

**Table 2.** Descriptive statistics of the respondents' households, Rupsa and Batiaghata sub-districts, Khulna.

| Variables | N = 200 | % | Variables | N = 200 | % |
|---|---|---|---|---|---|
| | | | Land ownership | | |
| **Rupsa** | 138 | 69 | Landless (0.02 ha) | 62 | 31 |
| **Batiaghata** | 62 | 31 | Marginal (0.02–0.2 ha) | 55 | 28 |
| | | | Small (0.2–1.0 ha) | 58 | 29 |
| | | | Medium (1.0–3.0 ha) | 20 | 10 |
| | | | Large (>3.0 ha) | 5 | 3 |
| **Gender** | | | House type | | |
| **Male** | 138 | 69 | *Kacha* (local materials) | 87 | 44 |
| **Female** | 62 | 31 | *Pucca* (brick-concrete) | 66 | 33 |
| | | | *Semi-pucca* (brick-tin) | 46 | 23 |
| **Age (years)** | | | Mobile phone access | | |
| **18–25** | 20 | 10 | Normal mobile phone | 171 | 85 |
| **26–40** | 72 | 36 | Smartphone | 109 | 54 |
| **41–60** | 77 | 39 | Mobile used in agriculture | 52 | 26 |
| **Above 60** | 31 | 16 | Interested in mobile app | 182 | 91 |
| **Education** | | | Drinking water source | | |
| **Illiterate** | 52 | 26 | Tubewell (>100 m) | 145 | 72 |
| **Primary ed.** | 56 | 28 | Tubewell (<100 m) | 40 | 20 |
| **Secondary ed. HSC and above** | 44 | 22 | Pipe supply | 6 | 3 |
| | 48 | 24 | Other (multiple) | 9 | 5 |

Access to Mobile Phones

Despite relatively widespread mobile phone usage in the study area, access to agriculture-related information via mobile phones was rare. Though 85% of the sampled farmers did use a personal mobile phone, most of these (74%) indicated that they did not use a mobile phone to access agricultural information. Farmers reported a number of reasons for the limited usage of mobile phones in agriculture. Major reasons were lack of ICT knowledge (81%), incompatible format and language (50%), lack of a smartphone (37%) and economic reasons (17%). Half of the farmers (54%) had access to a smartphone in their household. Younger farmers (10% of the sample) between the ages 18–25 years and middle-aged farmers between the ages 26-60 years (75% of the sample) were interested in a mobile phone application for receiving agricultural information. However, the illiterate farmers (9% of the sample) and farmers older than 60 (16% of the sample) expressed a preference for receiving agricultural information through face-to-face sources. Some of these older farmers (5% of the sample) observed that in-person communication was more reliable, since people could tell lies over mobile phones, especially regarding input and market prices.

*3.2. Crop Cultivation in Peri-Urban Khulna*

Agricultural practices differ by season. The agricultural calendar in Khulna is constituted by three crop seasons: khariff-I (pre-monsoon), khariff-II (monsoon) and rabi (winter). Khariff-I (mid-March to mid-June) is a transitional minor crop season. The two major crop seasons are khariff-II and rabi. Table 3 and Figure 3 present the crops cultivated in the study areas during these seasons. Paddy was the dominant crop of the majority of farmers. During khariff-II (mid-June to mid-November), it was cultivated by 71% of the farmers sampled and during rabi (mid-November to mid-March) by

about 33%. Focus group participants (n = 52) and agriculture extension experts (n = 20, see Table 1) reported that paddy was grown for household food security. However, they also reported that there had been a major shift in cropping practices over the past few decades, in response to hydro-climatic variability. Paddy farmers were increasingly growing different short-term crops and vegetables, such as gourds, beans, cucumbers, tomatoes and watermelons, alongside small-scale aquaculture. Agriculture extension officers in both Batiaghata and Rupsa confirmed the emergence of integrated farming systems spanning the three crop seasons. We also found a trend of increasing rainfall in the study area. This likely played a role in promoting cultivation of short-term vegetables integrated with aquaculture (see Supplementary Material A).

Extension officers observed that 10 to 20 years ago, farmers cultivated the local paddy varieties aus and aman. However, after introduction of improved paddy varieties (upsi) and high yielding varieties (HYV), farmers lost interest in aus paddy cultivation in khariff-I (mid-March to mid-June). At the time of our survey, the majority of farmers (71%) cultivated aman paddy only during khariff-II (see Table 2). Some farmers cultivated paddy with vegetables (18%), while others mixed vegetables with aquaculture (6%) or cultivated only vegetables (3%) during khariff-II. A few farmers (3%) said they left their land fallow during khariff-II, due to waterlogging and flooding.

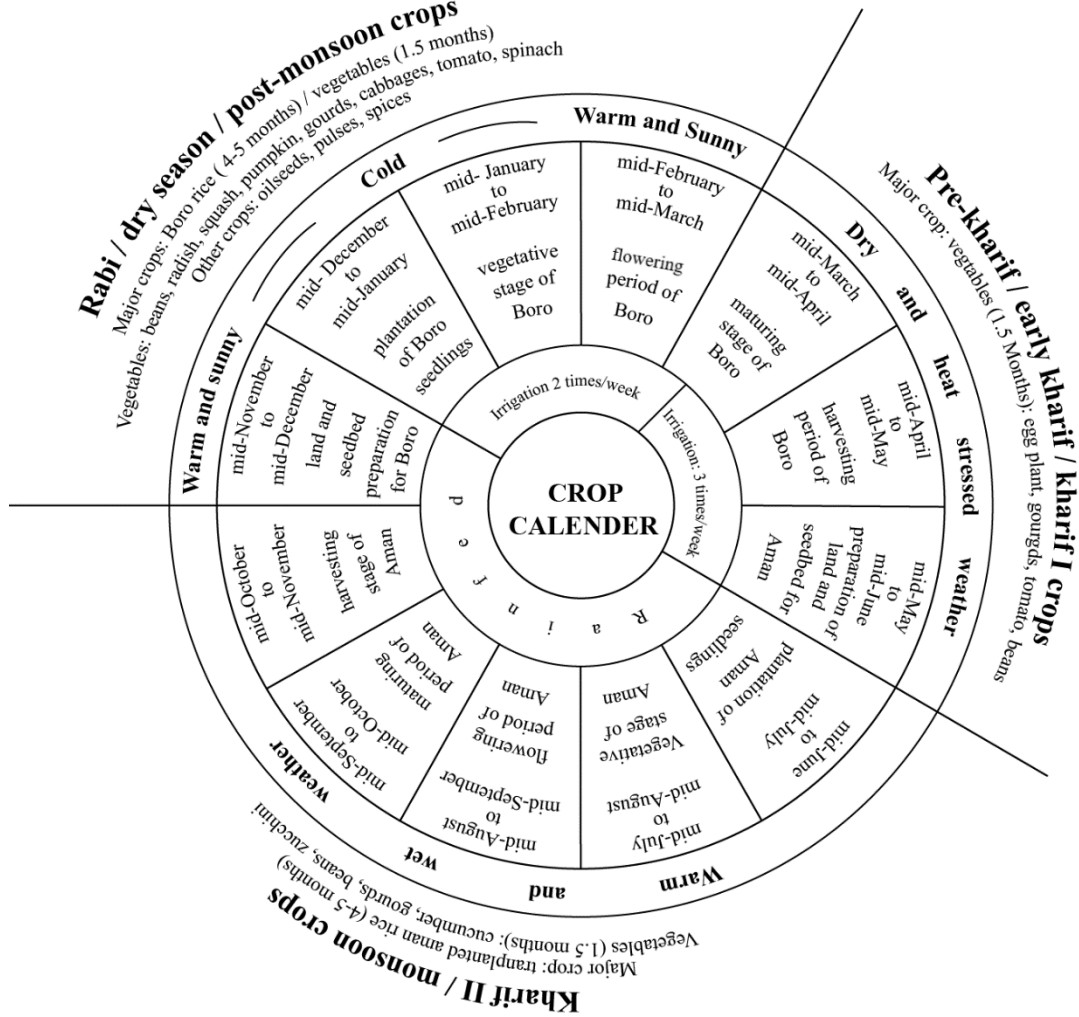

**Figure 3.** Crop calendar of peri-urban Khulna constructed based on group meetings and individual interviews.

**Table 3.** Cultivation practices among peri-urban farmers in Khulna during the three crop seasons

| Cultivation Practices | Khariff-I (Mid-Mar to Mid-June) | | Khariff-II (Mid-June to Mid-November) | | Rabi (Mid-November to Mid-March) | |
|---|---|---|---|---|---|---|
| | Frequency | % | Frequency | % | Frequency | % |
| Paddy | 47 | 24 | 141 | 71 | 66 | 33 |
| Paddy and other crop | 4 | 2 | 35 | 18 | 15 | 8 |
| Vegetables | 26 | 13 | 6 | 3 | 51 | 26 |
| Vegetables and other crop | 17 | 9 | - | | 5 | 3 |
| Aquaculture and Vegetables | 10 | 5 | 12 | 6 | 1 | 1 |
| Sesame and pulses | 2 | 1 | - | | 6 | 3 |
| Fallow | 94 | 47 | 6 | 3 | 56 | 28 |
| Total (N = 200 interviews) | 200 | 100 | 200 | 100 | 200 | 100 |

Note: due to rounding, some of the categories total 101%.

Rabi is the water-scarce dry crop season. *Boro* paddy (33%) and winter vegetables (26%) were the main rabi crops in the study area. Major vegetables grown during this season were cabbage, cauliflower, eggplant, spinach, beans, red amaranth, radish, pumpkin and tomato. Groundwater and surface water salinity increased during rabi. Farmers with irrigation facilities (33% of the sample) cultivated *boro* paddy during this time. Those with limited irrigation (26%) tended to cultivate short-term vegetables during rabi. Farmers without irrigation water left their lands fallow during rabi and khariff-I. Of the sampled farmers, 28% left their lands fallow in rabi and 47% did so in kharif-I. In rabi, about 8% of the sampled farmers cultivated paddy with vegetables, and 3% of farmers cultivated sesame as a short-term crop. Very few farmers mixed aquaculture with vegetable cultivation (0.5% of our sample) or vegetables with another short-term crop (2.5%) during rabi. However, in the low-lying beel areas on the outskirts of Khulna, aquaculture and aquaculture mixed with vegetables were popular year-round, practiced by 12% of the farmers surveyed.

### 3.3. Agricultural Information Sources and Their Value to Farmers

Agricultural information here refers broadly to information that plays a potential role in agricultural decision-making, such as seasonal weather forecasts, water availability, input prices and availability, crop selection, disease control and market prices. The farmers and experts interviewed or participating in focus group sessions for this research identified 18 agricultural information sources used by farmers in the study area. We classified these into five broad categories: informal contacts, formal contacts, education and training programs, traditional mass media and modern ICT tools/social media (Table 4).

We explored the value that farmers attached to each of these sources, and reasons for their preferences, again based on findings from the interviews and focus groups. First, from the farmer interviews, we determined the percentages of respondents favoring a particular information source. Then, we derived the value farmers attached to that source by the proportion of respondents favoring it. These values were subdivided into five categories: very high value (more than 80% of the respondents favored it), high value (60–80% of respondents favored it), medium value (40–60%), low value (20–40%) and very low value (<20%). We discerned reasons why the various sources were perceived as valuable (or not) from comments made by farmers in the interviews and focus groups. For example, 66% of the farmers interviewed (i.e., between 60–80%) indicated valuing information obtained from peers, while 100% of the interviewed farmers (i.e., >80%) indicated valuing information received from input dealers. These sources were thus determined to have a high and very high value, respectively, to local farmers. The farmers attached very low value (i.e., <20%) to information available from ICT platforms, such as mobile phones, the internet and social media. Below, we elaborate on these different types of information in more detail.

*Informal contacts.* This is the traditional way farmers communicate and access information. It is highly social, often requiring no extra time or added expenditure. Farmers said that personal experience gained over years, and consultation with peers, brought new insights regarding land preparation, crop and variety selection, water availability, seasonal weather and emergencies. Our respondents had 20 years of farming experience on average. Therefore, the majority had vast personal experience to draw on for agricultural decision-making. In addition to exchanges with peers, farmers gained information via informal contacts with trusted input dealers and local retailers, particularly regarding input prices, new crop varieties, disease control, cultivation methods and crop production.

While some farmers indicated obtaining information from extension officers, in practice, the majority went directly to trusted input dealers rather than to the extension office. According to one farmer interviewed, "Input dealers are always available and accessible, and they have all kinds of information for agricultural decision-making." Communication with input dealers was considered easy, efficient and well-aligned with farmers' information needs and time schedules.

*Formal contacts.* The farmers interviewed considered formal information sources to be of medium (40–60%) to very low (<20%) value. Medium value indicates that 40–60% of farmers favored the information source, and very low value indicates that <20% of farmers favored the source. The DAE, via the SAAOs, was the main formal source of information, for example, regarding seasonal crop cultivation, disease control, new technologies, organic agriculture, soil health and government subsidies for farmers. Yet, each SAAO was responsible for providing information to some 2000 farmers, and did this mainly through individual and group-based interactions. It was thus difficult to get sufficient information to all, within the needed timeframe. From the farmers' perspective, too, it was difficult to obtain information from their SAAO or even from the DAE office, as many farmers (79%) had secondary off-farm occupations during working hours. These prevented them from being able to access information through formal contacts during local extension officers working hours.

The Agriculture Information Service (AIS) is a governmental organization under the Ministry of Agriculture responsible for providing agriculture-related information to farmers and other interested stakeholders. However, none of the farmers participating in this research had used information from AIS. AIS employed both traditional media (mainly radio and television) and ICT-based platforms (the internet and social media) to disseminate information on agricultural production, technology and innovation.

Non-governmental organizations (NGOs) were occasionally active in providing agricultural information. Overall, however, farmers attached little value to information accessed this way. Similarly, the Union Digital Centre (UDC), a formal government entity established to provide information services at the union level, had seldom been accessed by farmers for agricultural information.

*Education and training programs.* Education and training were mostly provided by the DAE and development partners. Farmer Field Schools (FFS) was an approach often used by the DAE for group-based education and training [56,57]. Topics addressed at FFS events included seasonal crops and cultivation methods, disease outbreak awareness, new crops, variety advice and government subsidies. The interviewed farmers in Batiaghata attached high value to the information they gained from DAE FFS events. However, we found no farmers with FFS experience in Rupsa. This is because FFS events were typically conducted under specific government or non-government projects, meaning that they were unavailable outside the project localities. The DAE expressed the aim of its FFS events as to build farmer capacity, enabling farmers to take informed decisions in relation to crops and livelihoods.

Besides FFS events, the DAE also organized group discussions, yard meetings, and field demonstrations for education, training and awareness-raising. Farmers indicated that these programs had high to medium value in terms of agricultural information provision. At the time of this research, weather information was a key focus of DAE farmer training and awareness-raising. Its aim was to help farmers better adapt to hydro-climatic variability and livelihood vulnerabilities. Leaflets and brochures were being distributed as part of these efforts. Some 40–60% of the farmers interviewed had received these DAE leaflets or brochures. They attached medium value to them as an information source (Table 4).

*Traditional mass media.* Television, radio and newspapers remained an important source of agricultural information among the farmers interviewed. Television, particularly, was a preferred medium. Some 69% of the interviewed farmers attached high value to agricultural information obtained from television. However, few of the farmers in our sample used newspapers (9%), and just a fifth used radio (20%) as a source of agricultural information. Reading newspapers was not a common practice among farmers around Khulna. This may be due to lack of access or the high cost of newspapers, alongside the ready availability of mobile phones, television and the internet. Farmers participating in the focus groups said that radio was practically extinct as a source of agricultural information. However, a few aged farmers in the Batiaghata group indicated that during severe weather events they got weather updates from both radio and television. Since 1983, the television magazine show *Hridoye Mati O Manush* has been one of Bangladesh's most popular programs on agriculture. All of the farmers participating in the focus group sessions commented on the value of this program for gaining information about new crops, cultivation practices, technologies and innovation in agriculture. However, in practice, all of the farmers said they still based their decisions mainly on traditional information, and planned farming activities in line with their previous experiences and tradition. Thus, information from these media seemed to hardly influence farmers decision-making.

**Table 4.** Available sources of agriculture-related information and the value farmers attached to each for agricultural decision-making in peri-urban Khulna. Value categories are as follows: "very high", "high", "medium", "low" and "very low". These reflect, respectively, the percentages >80%, 60–80%, 40–60%, 20–40% and <20% of farmers favoring that source. Data were drawn from interview results and focus group discussions.

| Main Information Sources | | Current Value to Farmers (N = 200) | Reason the Source Was or Was Not Valuable to Farmers |
|---|---|---|---|
| **Informal Contacts** | Personal experiences | Very high (100%) | - Tried and true nature of personal expertise and skills<br>- Experience with same crops at same locality |
| | Consultation with peer farmers | High (66%) | - Easy to communicate<br>- Personal kinship and friendship<br>- Always available and accessible |
| | Input dealers, retailers and company representatives | Very high (100%) | - Easy to communicate<br>- Personal kinship ties<br>- Dependence on loans for inputs<br>- Warranty on input services<br>- Proactive information services<br>- Feedback mechanisms exist |
| **Formal Contacts** | DAE | Medium (44%) | - Difficult to communicate<br>- Time-consuming process<br>- Time schedules do not match |
| | AIS/AICC | Very low (2%) | - Difficult to communicate<br>- Limited access for farmers<br>- Traditional media dependent<br>- Limited service coverage<br>- No feedback and limited interaction on service provision |
| | Union Digital Centre (UDC) | Very low (5%) | - Limited expertise on agriculture<br>- Time consuming process<br>- Hardly useful to farmers |
| | NGOs | Very low (9%) | - Limited service delivery<br>- Project and beneficiary based<br>- Availability for limited periods |

**Table 4.** *Cont.*

| Main Information Sources | | Current Value to Farmers (N = 200) | Reason the Source Was or Was Not Valuable to Farmers |
|---|---|---|---|
| **Education and Training** | Farmer field schools | High (66%) | - Easy to communicate<br>- Group learning and sharing<br>- Technical knowledge improved<br>- Long-term skills learned |
| | Individual education and training | High (60%) | - Training on contemporary issues<br>- Builds capacity on new technologies for crop production |
| | Group meetings, field days, workshops and conferences | High (60%) | - Easy to communicate<br>- Shared learning process<br>- No extra time needed<br>- Personal kinship and social assets |
| | Extension materials and leaflets | Medium (44%) | - Easy to communicate<br>- Advanced information<br>- Free of cost<br>- Builds knowledge and awareness |
| | Fairs and exhibitions | Medium (40%) | - Face-to-face communication<br>- Fun to see others |
| **Traditional Media** | Newspapers | Very low (9%) | - Limited access at village level<br>- Limited reading culture<br>- Some farmers illiterate<br>- Limited information on agriculture |
| | Radio | Very low (18%) | - Very limited use by farmers<br>- Traditional technology<br>- Information not location-specific<br>- Information not time-specific |
| | Television | High (69%) | - Programme formats easy to follow<br>- Live programmes<br>- News programmes share innovation<br>- Many television channels |
| **ICT/Social Media** | Mobile calls, direct messages, multimedia messages | Very low (16%) | - Information not location specific<br>- Unnecessary messages from operator<br>- Lack of education and awareness<br>- No feedback mechanism exists |
| | Mobile phone applications | Very low (4%) | - Lack of ICT knowledge<br>- Lack of smartphone access<br>- Top down/not informed<br>- Lack of awareness of information<br>- No feedback mechanism exists |
| | The internet, websites, E-Krishi, E-Kiosk and social media | Very low (3%) | - Lack of ICT knowledge<br>- Lack of smartphone<br>- Lack of education and training<br>- High cost of internet facilities<br>- No feedback mechanism exists |

*Modern ICT tools and social media.* ICT applications in agriculture began to emerge in Bangladesh during the 1980s [2]. Yet, the farmers participating in this research attached little value to them. Mobile phone calls were valued by 26% of the interviewed farmers as an information source for agriculture, with smartphone applications valued by 4% and social media by 3%. Traditional sources were much more highly valued, particularly personal experience (valued by all of the interviewed farmers), consultation with peers (valued by 66%) and information from input dealers (valued by 100%). Almost all organizations active in the agricultural sector had modern ICT platforms, including websites, call centers, smartphone applications and social media platforms, which they used to share and

disseminate agricultural information. However, farmers attached little value to these for agricultural decision-making (see Table 4).

Farmers and experts expressed four key limitations to information uptake from ICT platforms: lack of location- and time-specific information, lack of accessibility, lack of awareness and lack of capacity. For example, in June 2012, AIS launched a mobile phone-based information platform called the *Krishi* Call Centre. However, none of our respondents used that platform. Other ICT platforms, both government- and non-government-run were similarly disregarded by farmers, who noted that they had little or no value for their own decision-making. Perhaps this is because these platforms were created in a mostly non-participatory and top-down manner. Interaction with farmers in their development was limited. Moreover, information communicated via these platforms tended to be out of date. Facebook was a popular social media platform in the study area, but it was not being used to share agricultural information. Training, capacity, participation and partnership efforts between information producers and users would be required to improve the overall local impact of the ICT platforms in the study area.

### 3.4. Hydro-Climatic Information Availability and Quality

Farming communities around Khulna are highly vulnerable to hydro-climatic variability. Major changes have been observed in, for example, monsoon rainfall patterns; river discharges and tidal characteristics; salinity intrusion in soils; groundwater and surface water; temperature stresses (droughts and hot and cold spells); and weather emergencies such as thunderstorms, hailstorms and cyclones. However, the extension experts (n = 7) noted a lack of active information provision to farmers on these topics. The BMD and BWDB were the two main government organizations providing hydro-climatic information at the national, regional and local levels. However, both farmers and extension experts indicated the inadequacy of the BMD and BWDB information for use and understanding by farmers. Local BMD officials said that BMD and BWDB divisional and local offices had no obligation to provide information services to farmers. The BMD collected weather information, such as temperatures, rainfall, wind speeds, wind directions and hours of sunshine, from local weather stations and transmitted it to it national headquarters. Likewise, BWDB collected water information, such as river discharge, water levels and salinity, through local stations and sent it on to their national headquarters. The collected information was then disseminated at the national level through traditional and ICT platforms. In this process intermediaries and end-users had no role in designing and delivering the information. Similarly, BWDB's Flood Forecasting and Warning Centre (FFWC) provided flood forecasts and warnings pertaining to all major rivers, but with limited access, comprehensibility and actionability at the community and individual farmer levels.

We analyzed farmers' information access behavior particularly for hydro-climatic information from different sources. Results show that 69% of the 200 farmers in our sample obtained weather information from television, and 67% from peers (Figure 4A). About 22% of the farmers also inquired at their extension field offices regarding upcoming weather hazards, and 20% of farmers listened to the radio for weather updates only during bad or extreme weather. Although about 85% of the farmers in our sample had mobile phone access, only 15% used mobile phones to access hydro-climatic information, either from peer farmers or from the agricultural extension department. Very few farmers (9%) read the newspaper for weather updates, and only 8% said they were experienced in understanding local weather phenomena. Nevertheless, during emergency weather events, farmers did obtain information from multiple sources, including the agriculture extension department, television and radio, local and national newspapers, input dealers, NGOs and community organizations.

According to the farmers in our sample, the overall quality of the hydro-climatic information available was unsatisfactory. For example, it was not sufficiently location- and time-specific to help them to make specific decisions. The farmers in the focus groups, too, said that they could not rely much on the hydro-climatic information broadcast via radio and television. They added that only in about 5–10% of instances was the information somewhat applicable to their location. Farmers also noted the lack of crop advisory services related to hydro-climatic forecasts, which often came late.

More than 60% of the respondents said that the existing weather information services were poor or very poor at the local level and of insufficient quality to aid in agricultural decision-making (Figure 4B).

Some 35% of the farmers indicated that the present information quality was acceptable, especially during emergency weather events. They observed that emergency forecasts on cyclones, storm surges and floods were disseminated proactively by governmental, non-governmental and community organizations. Based on that information, farmers could sometimes prepare in advance for extreme weather events. Farmers and experts noted that field crops were difficult to protect during extreme weather events. However, if sufficient information was available in advance, harvested and mature field crops, livestock, fishponds and household and farm assets could be protected from major damage.

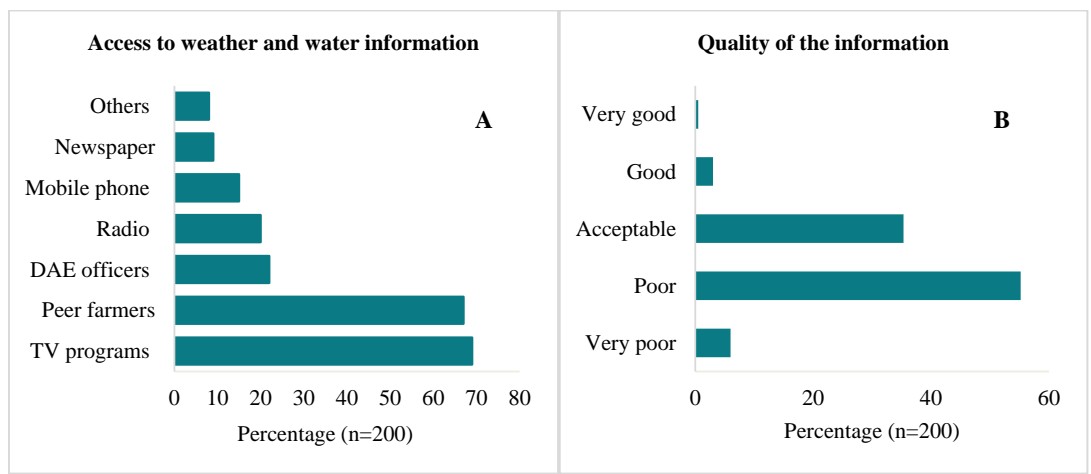

**Figure 4.** Farmers' access to hydro-climatic information sources and their perception of the quality of information from those sources. (**A**) shows farmers' access to hydro-climatic information from different sources, and (**B**) shows an overall judgement about the hydro-climatic information quality found from all sources.

### 3.5. How Farmers Used Available Information in Decision-Making

Farming communities in Khulna considered hydro-climate hazards "God-given phenomena". Thus, their information-seeking behavior for climate events was limited, and their decision-making was driven mainly by tradition and personal experience. However, within our sample, young farmers (10%), educated farmers (46%) and farmers with diversified operations (about 17%, 24% and 14%; see Table 4) reported using multiple information sources, especially for crop selection, planting times, fertilizer and pesticide application, irrigation management, harvest planning and marketing [58]. Consideration of hydro-climatic information for crop management decisions varied significantly between the farmer categories. For example, smallholder paddy farmers took key decisions based mainly on their personal experience and traditional practices. But farmers with integrated operations were more apt to try to hone their decisions in consultation with peers and advice from input dealers and extension officers.

The focus group discussions confirmed that farmer age, gender, education, farm type, farm size, personal beliefs and risk-aversion were important characteristics affecting the amount of information consulted in decision-making. For example, young, educated male farmers frequently communicated with the field agriculture officers and input dealers for informed decision-making. Whereas female farmers hardly communicated with the field agriculture officers and input dealers for agricultural decision-making. This may be because women farmers were less engaged in field crop cultivation. The majority based their agricultural decisions on their own previous experiences and advice from peer farmers nearby. Routine use of hydro-climatic information was not commonplace among the sampled farmers. It was particularly uncommon among smallholders (79% of the sample); they often also had off-farm income sources in or around the city.

Five key determinants of cropping plan decision-making were identified: agronomic considerations, economic considerations, resource-related considerations, farmland-related considerations and climatic considerations. In our study area, we identified two additional determinants of agricultural decision-making: individual considerations and input support from government and non-governmental agencies. Table 5 presents the expanded list of key determinants of cropping plan decision-making. Smallholder farmers around Khulna grew paddy due to individual considerations: they wanted to ensure their household's food security. As one farmer said, "If we have rice at home, we are free from the stress of buying rice at the market." Furthermore, a good yield in the previous season informed farmers' cropping decisions in the next seasons. Due to the increasing frequency of hydro-climatic events, farmers indicated that they now preferred short-term, salt- and drought-tolerant varieties. All farmers perceived hydro-climatic variability as high in Khulna; and farmers always sought to make better decisions based on their growing experience and perception of hydro-climatic risks and uncertainties.

Figure 5 presents perceptions of hydro-climatic variability among the sampled farmers. Additionally, the farmer focus groups sketched 14 major farming decision areas: crop selection, land preparation, variety selection, seeding, hiring labor, transplantation, weeding, irrigation, fertilizer application, pesticide application, harvesting, processing and storage, seed preservation and marketing. These represent key tactical, or medium-term decisions, that can be aided by multiple types of hydro-climatic information. For example, farmers might decide to purchase particular seeds, inputs or varieties based on hydro-climatic forecasts. Adaptive decisions, in contrast, refer to long-term strategic decisions aimed at adjusting to hydro-climatic variability and change. For example, paddy farmers at Batiaghata had switched from paddy to watermelon, because of the scarcity of irrigation water and problems of salinity intrusion during the *boro* paddy season. Decisions such as these drew on information from different sources and farmers' risk perceptions drawing on traditional knowledge and experience.

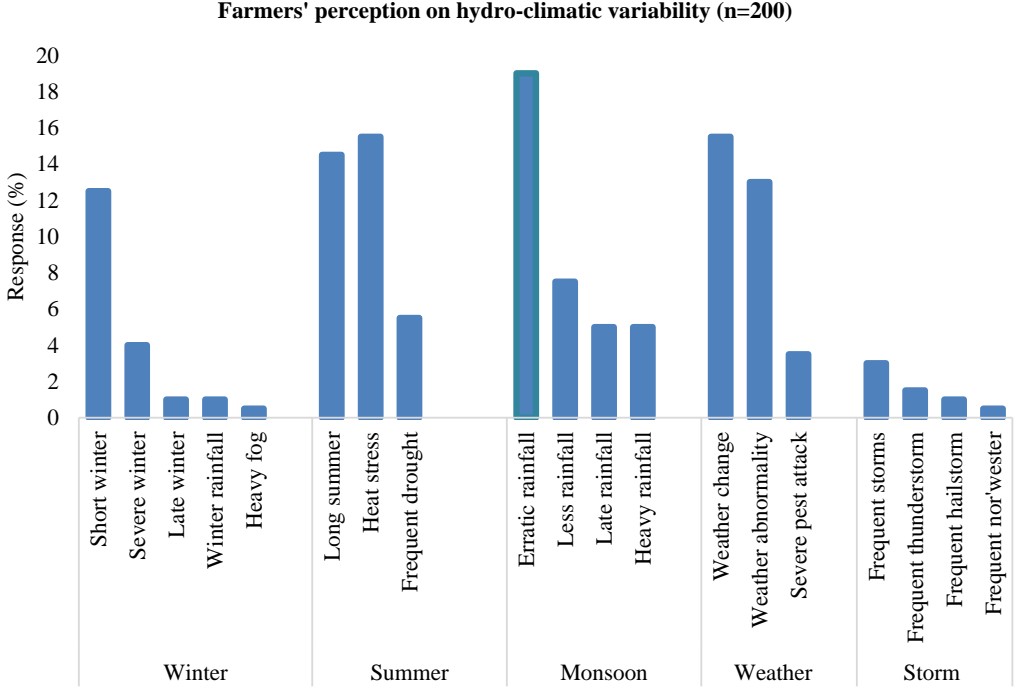

**Figure 5.** Farmers' perceptions of hydro-climatic variability in the Bengal Delta around Khulna, Bangladesh. Bar graphs show farmers' perceptions of hydro-climatic variability during three crop seasons (winter/ kharif-I, summer/kharif-II and monsoon/rabi) and changes in overall weather and storm characteristics in Khulna. To derive these, we asked farmers what were the major water and weather hazards for crops. Farmers revealed that a short winter, long summer, erratic rainfall and temperature stress were the key variabilities affecting local crops and agricultural decision-making. They also reported hydro-climatic variability in terms of weather abnormality, severe pest attacks and storm frequency.

**Table 5.** Key determinants of cropping plan decision-making, after Dury, Garcia, Reynaud and Bergez [59].

| Determinants | Sub-Groups | Description |
|---|---|---|
| **Individual Considerations** | - Personal preferences<br>- Food security<br>- Tradition<br>- Land ownership | - Personally motivated choice<br>- Food security/food stock at home<br>- Tradition/practices from ancestors<br>- Land ownership status |
| **Agronomic Considerations** | -Crop characteristics<br><br><br>- Soil quality<br><br>- Management facilities | - Yield performance in previous year(s)<br>- Crop cycle and time to harvest the crop<br>- Pest and disease outbreak/sensitivity<br>- Soil salinity and soil quality<br>- Land availability for cultivation<br>- Soil water content/moisture<br>- Field operations/activities<br>- Irrigation facility<br>- Fertilization<br>- Weed, pest and disease control<br>- Crop harvest/collection<br>- Storage and preservation |
| **Economic Considerations** | - Profit<br>- Cost of production<br>- Price uncertainty<br>- Demand uncertainty | - Profit margin high/medium/low<br>- Input costs and availability<br>- Labor cost and availability<br>- Easy to sell from field to market<br>- Crop price uncertainty/fluctuation<br>- Crop demand uncertainty/fluctuation |
| **Resource-Related Considerations** | - Labor<br>- Water<br>- Technology<br>- Finances<br>- Scope for loan | - Labor availability<br>- Water availability and irrigation facility<br>- Technology support and availability<br>- Financial situation/capacity<br>- Loan opportunity (money/input) |
| **Farmland-Related Considerations** | - Management<br>- Spatial<br><br>- Land suitability | - Nearby or remote field distance<br>- Transport facility to crop field<br>- Location and accessibility of crop field<br>- Land type (high/low/waterlogged) |
| **Climatic Considerations** | - Seasonal and short-term weather conditions<br><br><br><br>- Traditional knowledge for weather predictions | - Seasonal weather conditions<br><br>- Weather hazards (fog, cold spell, hail, flood, drought, hot spell and disease outbreak)<br>- Perception of rainy season<br>- Perception of summer season<br><br>- Perception of water stress and water availability |
| **Support from Outside Actors** | - National<br>- Development agencies<br>- NGOs | - Input support from DAE<br>- Input support from NGOs<br>- Education and training on new technology and cultivation practices |

*3.6. Farmers' Adaptation to Water Stresses and the Role of Information*

Both the farmers and the experts consulted attached a high value to the role of information in agricultural adaptation practices. All the interviewed farmers perceived hydro-climate patterns as having rapidly changed over the past 10–20 years (see Figure 5). One farmer interviewed in the village of Raingamari claimed that he had not gotten satisfactory production from his paddy fields the previous five years due to hydro-climatic hazards. "Now I work as a labourer instead of a sharecropper," he said. Another farmer in the same village agreed, "Rainfall does not follow the traditional rules and characteristics, and that impacts crop production and crop-related decision-making." A farmer in the village of Jharbhanga blamed human

activities for the increased weather stress, "Weather is now polluted by human activities . . . . [N]ature is taking revenge on us through frequent weather events."

Farmers had few response options for field crops, especially during extreme events like cyclones, storm surges, heavy rains, hailstorms, droughts and floods. Farmers and experts indicated that if these hydro-climatic events could be forecasted and communicated farther in advance with sufficient lead-time (between 1 to 2 weeks), they could at least safeguard harvested and mature crops, livestock and farm assets. Farmers were adapting some of their management practices. Already they had increased irrigation frequency to reduce the impact of drought stress and advanced pesticide applications to limit disease outbreaks during cold spells. Farmers had also adjusted their planting times to accommodate perceived hydro-climatic changes. Hydro-climatic information services could help farmers make these decisions. Such services could also alert farmers to the need for action to limit crop damages from extreme weather. Untimely rain or irregular rain, a short winter season, an early or late start of the rainy season, a prolonged rainy season, heavy rain, floods, waterlogging, salinity intrusion and irrigation water scarcity were the major hydro-climatic stresses reported by farmers in interviews and focus group discussions. Similarly, experts reported hydro-climatic variability in the study area, suggesting that localized and reliable information services could potentially improve local farmers' adaptation practices.

To avoid water stress in field crops, farmers in Khulna sometimes irrigated with mildly saline water. In Batiaghata, farmers reported negotiating with the local sluice operator to let in water from the river while its salinity was still relatively high, before the start of monsoon season. Batiaghata farmers (62 of the 200 in our sample) indicated that saltwater from the river pushed through the freshwater canal, bringing the freshwater to their field more quickly. Before the saltwater reached their location, the sluice gates were closed to avoid saltwater intrusion into the *boro* rice fields. Farmers also applied extra chemical gypsum fertilizer to reduce the effects of salinity on field crops. Furthermore, the construction of ponds and reservoirs had become common practice around Khulna, to meet dry season irrigation demands and as an adaptation strategy to reduce water stress. Integrated farming was on the rise as well, often combining rice, fish, vegetables and livestock. This too helped farmers adapt to increasing salinity and hydro-climate stress. Some 14–24% of the interviewed farmers had integrated farming systems (see Table 3).

Farmers in peri-urban Khulna had access to both formal and informal sources of information on production technologies, inputs, disease control and ways to cope with and adapt to climate change. However, they still lacked access to hydro-climatic information services for agricultural decision-making. According to the interviewed farmers and experts, hydro-climatic information services in the study area were fragmented, top-down and not actionable. For example, most farmers did not understand the probabilistic forecasts of BMD and BWDB, and many were unable to apply available forecasts in decision-making. Much of the forecast information was in English, and it often came too late to be of use. Farmers, extension officers and experts alike were unaware of the existence of the BMD smartphone weather application, though the tool was available in the study area. Similarly, flood forecasts from FFWC generally did not reach farmers through extension channels. Soil and water salinity information was similarly unavailable to farmers.

SAAOs provided agricultural extension services, but had limited interaction with farmers individually. Furthermore, they too had limited knowledge of hydro-climatic forecasts, and thus played no role in sharing and disseminating these to farmers. One way to increase the reach of such forecasts would be to increase extension agents' affinity with them. Furthermore, extension agents could be called upon to advise on the design of forecasts and information services, to ensure their greater applicability at the local level.

## 4. Discussion

This study aimed to address three research questions regarding (1) information availability, (2) perceived quality and (3) the role of hydro-climatic information for farmers in the Bangladesh delta.

To answer these questions, we adopted an exploratory research framework combining participatory tools with analysis of climate data.

### 4.1. Information Availability and Farmer Preferences

Our results indicate that peri-urban farmers in the Bengal Delta have access to agricultural information from five main sources: informal contacts, formal contacts, education and training programs, traditional media and ICT/social media platforms. Farmers mainly rely on informal contacts with peers, dealers and retailers for agriculture-related information [27,60,61]. Interestingly, Miah et al. [60] found a low preference for group-based sharing among fish farmers in the Muktagacha sub-district of Bangladesh, whereas we found a high preference for group-based sharing of agricultural information by farmers. Extension officers view group sharing as an effective means to disseminate information through which they can reach a large number of farmers.

Although the majority of farmers depended on informal contacts as their main source for all kinds of information, specific and effective information comes primarily through formal extension departments. Hydro-climatic information, on extremes, seasonal weather forecasts, onset and ending of monsoon season, is hardly available to farmers via agricultural extension channels. This indicates a need to build a capacity of both farmers and extension agents to improve the communications and use of hydro-climatic information. In addition, local input dealers could be trained to provide information, as farmers attached a very high value to information coming from them. Input dealers expressed a keen interest in links with climate information services, the DAE and the weather department (BMD).

In addition, a common platform of the hydro-meteorological organizations could enhance sharing and disseminating locally specific hydro-climatic information with and for farmers. This could also enhance the quality of extension services making a direct link with local hydro-climatic information. Currently, these organizations are highly fragmented in terms of services and have no mandate to disseminate the collected information at the local level.

### 4.2. Perceived Quality and Role of the Information Available

Farmers perceived the quality of the available hydro-climatic information to be poor, insufficiently specific and often too late to be of use in local agriculture-related decision-making (see Figure 4B). Many farmers in our focus groups and interview sample expressed distrust of television and radio weather forecasts, indicating they were not specific enough to meet their needs [17]. Extension officers and other experts expressed similar doubts about the credibility of the available information and its relevance for the study area. As a result, farmers base most of their decisions on their own experiences and informal contacts when confronted with weather related risks and uncertainties. Currently, farmers were only partially able to respond to situations related to extreme events. They emphasized that they need more location- and lead-time-specific forecast information for tactical decision-making and to safeguard their crops, livestock and farm assets.

Our findings show that most farmers in the study area are potentially vulnerable to increasing hydro-climatic variability (see Figure 5). They often depend on small fields for food and livelihood security (see Table 2). Previous studies have shown that vulnerability to hydro-climatic variability and reduced crop yields is affecting all three crop seasons in the study area [55]. Farmers have already modified a range of cropping practices to adapt to increased hydro-climatic variability [27,55,62]. However, the role of hydro-climatic information in agricultural decision-making is still limited, and most decisions are still driven by personal experiences and tradition [25,61,63]. More educated and advanced farmers are, however, making greater use of hydro-climatic information services. This indicates that capacity building could be used to enable farmers with little or no formal education to take advantage of hydro-climatic information services in agricultural decision-making [63–67].

### 4.3. Use of ICT in Agricultural Decision-Making

An important question is whether and how ICT-led information services could improve extension efficiency and service coverage. Our results show that currently only a few farmers are using ICT tools to access agricultural and hydro-climatic information. The main barriers identified were lack of ICT knowledge (81%), lack of smartphones (37%), limited understanding of the English language (50%), and poor timeliness and low perceived quality of the available information (61%). In addition, none of the interviewed farmers and only some of the extension officers were aware of existing ICT platforms as a potential source of hydro-climatic information. This is a common problem among information services designed and launched with little or no contact with end-users [37,68–71]. Newly developed ICT platforms should take into account the identified barriers and ensure that platforms are known by intermediaries and the identified end-users. To achieve this there is a need for stakeholder engagement to improve information uptake and mutual learning and co-production of information services [67,70,72]. In addition, there might be potential to overcome some of the identified barriers by developing appropriately designed ICT-based services in the local language, combined with capacity building of DAE officers and farmers. Also, the increasing availability of low-cost smartphones and location-specific tailored information creates new opportunities for developing web- and app-based services.

### 4.4. Recommendations for Tailoring Hydro-Climatic Information

The results of this study point to six recommendations for tailoring hydro-climatic information for farmers. First, information platforms should use the local language and provide a local interpretation of hydro-climatic forecasts [73].

Second, the location and time specificity and trustworthiness of forecasts should be prioritized [17,68,74,75].

Third, frequent interaction with farmers, via relevant training and awareness-raising programs, such as farmer field schools, should be used to develop farmers' understanding and planning culture, alongside forecast information tailored to the study area. The AIS, in cooperation with the DAE, could play a leading role in running farmer field schools and be explicitly mandated to provide hydro-climatic information along with their existing extension services to farmers.

Fourth, greater resources need to be made available for local extension services. ICT-led platforms such as smartphones are one way to effectuate an increased reach of information and extension recommendations, given a large number of farmers are still unserved for such information services. After all, 85% of the surveyed farmers already had access to mobile phones and 54% had access to a smartphone in their household.

Fifth, a detailed needs assessment is recommended to tailor hydro-climatic information services to the various groups of farmers in the study area. Information needs vary across different types of farmers, crop seasons and farming systems. Further, farmers require capacity to express their information needs, particularly where they are unfamiliar with such information services and have limited academic education background [76–81].

Finally, agricultural advisories should be designed using a co-production approach with and for local farmers, based on forecast information and local conditions. Co-production can lead to the creation of more usable weather advisory services and forecasts through ICT platforms, while helping to address challenges inherent in ICT-led information provision, such as equity, access, social acceptability and contextualization.

From this study, we can conclude that there is potential for hydro-climatic information services which are better tailored to the needs of the local farmers. ICT platforms such as smartphones and social media could play an increasingly vital role in tailored information exchange and communication with local farmers, to help them make climate-sensitive decisions. The current study identified 14 climate-sensitive farm decisions for which farmers and experts attributed a high value when it comes to the role that information services play, in response to a rapidly changing hydro-climatic environment, in terms of frequent variability. On a critical note, this paper also highlighted the added value that

personal contact over experiences and traditional practices has for farmers. A key challenge will thus be to embed the development and introduction of any information services in a process of interaction that contextualizes the new information, thus gaining trust and facilitating the integration of scientific forecasts into daily decision-making practices.

**Supplementary Materials:** The following are available online at http://www.mdpi.com/2071-1050/12/16/6598/s1, Supplementary Material A: Farmers' Interview Form, Supplementary Material B: Focus Group Discussion Form, Supplementary Material C: Expert Interview Form, Supplementary Material D: Climate Data Analysis.

**Author Contributions:** U.K. conducted this study supervised by S.W., F.L., D.K.D. and L.P.H. The supervisors contributed substantially in the study design, editing and commenting on article drafts for several rounds. S.R. and S.A. contributed to this article through extensive field work, data acquisition and synthesis of field notes for this study. All authors have read and agreed to the published version of the manuscript.

**Funding:** This research is an output of the "Waterapps—Water Information Services for Peri-urban Agriculture" project funded by the Netherlands Organization of Scientific Research (NWO) under its Urbanising Deltas of the World (UDW) program.

**Acknowledgments:** We are highly indebted to funders, partners, project coordinator Erik van Slobbe and the study area community for their contributions. We also thank Rashed Jalal for his kind assistance in preparing a map for this study. We would also like to acknowledge our colleagues Vincent Linden and Michelle Luijben for proofreading the article and Spyros Paparrizos for helping in climate data analysis and providing feedback on the article. Finally, we acknowledge the reviewers' comments and suggestions, while confirming that the authors of this study have no conflict of interests.

**Conflicts of Interest:** The authors declare no conflict of interest.

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
