# Peer review of "Role of Information in Farmers’ Response to Weather and Water Related Stresses in the Lower Bengal Delta, Bangladesh"

_sustainability, doi:10.3390/su12166598_

Round 1
Reviewer 1 Report
- Try to converge title, hypothesis and focus on results. Either title or results need a bit modifications.
- Introduction needs improvement in context of problem and significance of the study.
- How credible information user benefited with better decision support and outcome and what happened for using poor source can be sited in results section.
- How sectors' collaboration can increase access to credible information needs to be highlighted with diagram.
- Discussion section deserve a critical debate in light of the other studies.
- However, the findings are interesting and bear significance for the reader.
- Detailed comments can be found in attachment.
Author Response
Please see the attachment https://susy.mdpi.com/user/manuscript/open_review/69638f66519e5b89a685af3a152df587

Reviewer 2 Report
Congratulations on an interesting, important and well-written paper. With a strong methodology, detailed analysis and clear recommendations, this paper has the potential for influencing change if appropriate pathways to informing policy are in place or can be developed. You might like to consider the preparation of a policy brief to support that process.
Given the importance of the term hydro-climate to this paper, it would be useful to discuss the definition as it relates to this paper.
One or two minor suggestions are picked up in the attached PDF.
Good luck with the paper and your further efforts.
